# How Theoretical Evaluations Can Generate Guidelines for Designing/Engineering Metalloproteins with Desired Metal Affinity and Selectivity

**DOI:** 10.3390/molecules28010249

**Published:** 2022-12-28

**Authors:** Todor Dudev

**Affiliations:** Faculty of Chemistry and Pharmacy, Sofia University, 1164 Sofia, Bulgaria; t.dudev@chem.uni-sofia.bg

**Keywords:** metalloproteins, metal affinity and selectivity, dft calculations, pcm computations, molecular modeling

## Abstract

Almost half of all known proteins contain metal co-factors. Crucial for the flawless performance of a metalloprotein is the selection with high fidelity of the cognate metal cation from the surrounding biological fluids. Therefore, elucidating the factors controlling the metal binding and selectivity in metalloproteins is of particular significance. The knowledge thus acquired not only contributes to better understanding of the intimate mechanism of these events but, also, significantly enriches the researcher’s toolbox that could be used in designing/engineering novel metalloprotein structures with pre-programmed properties. A powerful tool in aid of deciphering the physical principles behind the processes of metal recognition and selectivity is theoretical modeling of metal-containing biological structures. This review summarizes recent findings in the field with an emphasis on elucidating the major factors governing these processes. The results from theoretical evaluations are discussed. It is the hope that the physical principles evaluated can serve as guidelines in designing/engineering of novel metalloproteins of interest to both science and industry.

## 1. Introduction

Metal cations are an indispensable and integral part of many proteins which confer on the host entities unique properties/functions ranging from protein structure stabilization to enzyme catalysis, signal transduction, gene expression, hormone secretion and respiration. Two dozen metal species, so-called biogenic metals, have been selected in the course of evolution to participate in biological processes in living organisms. Among them, the most frequently found metal cations are Mg^2+^, Ca^2+^, Zn^2+^, Mn^2+^, Na^+^, K^+^, Fe^2+/3+^, Co^2+/3+^ and Cu^+/2+^.

For the proper functioning of metalloproteins, highly reliable selection of the (native) metal cation from the intracellular/extracellular fluids is critical. Metalloproteins have adopted various strategies to selectively sequester the appropriate metal cation from the surrounding milieu as described in a number of literature sources [1,2,3,4,5,6,7,8,9,10,11,12]. Most often, it is the protein itself that regulates the selectivity process by creating a binding site cavity with suitable geometry, amino acid composition, rigidity and solvent exposure. In some cases, however, the cellular machinery steps forward and orchestrates the selectivity process by maintaining proper concentration of the competing metal species, thus favoring the native one.

Knowing the physical factors that govern the processes of metal binding and selectivity in biological systems is of particular importance as this not only deepens our understanding about the intimate mechanism of these events but, also, significantly enriches the researcher’s toolbox that could be used in manipulating/engineering novel metalloprotein structures with desired properties. Especially suited for elucidating the basic factors behind the metal selectivity in proteins are theoretical modeling studies mostly employing quantum-chemical (QM) calculations combined with polarizable continuum model (PCM) computations. Usually, the QM/PCM calculations are based on realistic models of the metal binding site including ligands from the first, second and, sometimes, third coordination sphere. There are several justifications for employing such an approach in studying the processes of metal binding and selectivity in metalloproteins. First of all, the interactions between the metal cation and its first-/second-shell ligand surroundings are strong (especially with doubly and triply charged metal cations) and of electrostatic origin. These are fully characterized by QM calculations (most recently as density functional theory (DFT) computations) as they account explicitly for the entire set of electronic effects involved (including the charge transfer to/from the metal cation and polarization of the participating entities). Furthermore, the weaker interactions in the system (due to hydrogen bonding and/or van der Waals contacts between protein and/or water ligands) are also very efficiently treated by modern post-Hartree–Fock (HF) and DFT methods. The relatively small size of the system under study allows for employing high-level sophisticated QM methods and basis sets which, together, enhances the precision and reliability of the results obtained. Note that the metal first-shell/second-shell ligand interactions dominate the energetics of the metal binding site as the strength of the metal-ligand interactions quickly fades away with the distance and, accordingly, the contribution from the more distant protein ligands to the overall metal-protein interaction energy becomes fairly negligible [13]. Thus, the effect of the bulky protein matrix on the metal binding site affinity/selectivity can be safely modeled as a continuum dielectric by PCM calculations where a particular dielectric constant between 4 and 78 (4—deeply buried site, 29—relatively solvent accessible binding pocket, to 78—fully solvent-exposed site) is assigned to the binding cavity. Notably, the combination between DFT calculations and PCM computations provide an accurate picture of the thermodynamics of the metal ion competition in these systems, as all the effects accompanying the process are accurately and reliably treated. Generally, the QM/PCM calculations do not aim at reproducing the absolute values of the metal binding energy/free energy of the protein but, instead, are destined to provide dependable trends of changes in thermodynamic quantities with differing variables of internal or external origin.

In this review, we aim to summarize recent findings in the field of metal binding and selectivity in metalloproteins with a focus on elucidating the major factors governing these processes. The results from theoretical evaluations, mostly from our group, will be discussed. It is the hope that the physical principles evaluated can be employed as guidelines in the designing/engineering of novel metalloproteins of interest to both science and industry. The review is organized in the following manner: each section is dedicated to a given factor influencing the metal selectivity. Its role will be illustrated with one (or more) example(s) for a particular protein system. The implications of theoretical findings on the protein biochemistry will be discussed. Details for the computational approaches/techniques employed are not given here, as the original articles are relied on to provide the necessary information.

## 2. Intrinsic Physicochemical Properties of the Metal Cations Determine to a Great Extent the Metal Selectivity of the Binding Site

### 2.1. Competition between Ca^2+^ and Sr^2+^ in Calcium Receptors/Signaling Proteins

Strontium (Sr^2+^), a member of the alkaline earth metal group of the periodic table, is employed in medicine as a diagnostic or therapeutic agent. Its radioactive isotopes ^85^Sr and ^89^Sr—with a half-life of ~65 and ~51 days, respectively—are used to follow the Ca^2+^ kinetics and treat bone cancer sufferers [14,15]. Another strontium formulation, strontium ranelate, containing stable, non-radioactive strontium isotopes, has been shown to exert beneficial effects in treating people with osteoporosis, mostly postmenopausal women [14,16,17,18,19]. It has been shown that strontium exerts a dual beneficiary effect: it reduces the bone degradation and promotes bone anabolism [14,16,17,18,19].

Strontium’s clinical applications stem from its ability to closely imitate biogenic calcium ions (Ca^2+^) and follow Ca^2+^-specific pathways involved in cell signaling and bone formation. Both Ca^2+^ and Sr^2+^ are spherical, doubly charged “hard” cations that strongly prefer “hard” oxygen-containing ligands (side chains of Asp^−^/Glu^−^, Ser, Asn/Gln and backbone peptide groups). Their ionic radii are similar: 1.0/1.06 Å for Ca^2+^ and 1.18/1.21 Å for Sr^2+^ in hexacoordinated/heptacoordinated complexes, respectively [20]. The respective hydration free energies are also quite close: −359.7 kcal/mol for Ca^2+^ and −329.8 kcal/mol for Sr^2+^ [21]. In the human body, the two metals behave similarly, both exhibiting distinct bone-seeking properties [14]. They possess flexible coordination/hydration spheres comprising 6 to 9 ligands [22,23]. As a Ca^2+^-mimetic species, Sr^2+^ radio-isotopes preferentially accumulate at sites of increased osteogenesis, thus focusing the radiation exposure on the cancerous regions. By mimicking Ca^2+^, non-radioactive Sr^2+^ has been postulated to bind and activate the calcium-sensing receptor (CaSR), a representative of the G-protein coupled receptor (GPCR) family [14,16,19,24,25]. CaSR activation triggers a cascade of signaling pathways promoting apoptosis of bone tissue-degrading cells (osteoclasts) and differentiation of bone-synthesizing cells (osteoblasts). In addition to CaSr, Sr^2+^ can also compete with Ca^2+^ in binding to proteins such as parvalbumin, alkaline phosphatase, calbindin, Ca^2+^-sensitive ATPase, and calmodulin [26,27].

The intimate mechanism of Sr^2+^ activation of CaSR is, however, not fully understood. Several important questions remain: How Ca^2+^/Sr^2+^-selective are the metal binding sites of the activated CaSR? How efficiently could the “alien” Sr^2+^ compete with the native Ca^2+^ for binding to the receptor? What are the key determinants of the metal affinity/selectivity of CaSR in the activated state? To address these questions, a QM/PCM modeling study has been undertaken [28] and the Gibbs free energies for the Ca^2+^ → Sr^2+^ exchange in different dielectric media have been evaluated, as shown in Equation (1):[Sr^2+^-aq] + [Ca^2+^-CaSR] → [Sr^2+^-CaSR] + [Ca^2+^-aq](1)

In Equation (1), [Ca^2+^/Sr^2+^-CaSR] and [Ca^2+^/Sr^2+^-aq] represent the metal cation bound to receptor ligands inside the binding pockets and unbound outside the binding cavity (in bulk solvent), respectively. A positive free energy for Equation (1) implies a Ca^2+^-selective site, whereas a negative value suggests a Sr^2+^-selective one.

The Ca^2+^/Sr^2+^-loaded binding pockets of CaSR (Sites 1–4 [19]) have been modeled and their thermodynamic characteristics evaluated [28]. Site 1 is situated in a loop region where backbone peptide groups of Ile81, Ser84, Leu87 and Leu88 orbit the metal cation. The metal center in Site 2 is directly coordinated by the side chain of Thr100 and indirectly (via a water molecule) by the side chain of Asn102. The calcium ion in Site 3 is bound in an outer-shell mode (via water molecules) to Ser302 and Ser303, whereas the side chain of Asp234 and backbone carbonyl groups of Glu231 and Gly557 coordinate the metal cation in Site 4 in an inner-shell fashion. The role of bound Ca^2+^ ions in activating CaSR has been found to be mostly structural: they stabilize the active state by strengthening the homodimer interactions between membrane-proximal domains [19].

The modeling study reveals that the metal binding sites—although comprising a different number and type of protein ligands, overall structure and charge state—are all selective for Ca^2+^ over Sr^2+^ (Figure 1). Thus, strontium is predicted to be unable to dislodge the cognate calcium from the respective metal centers. The four binding sites, regardless of their structural differences, exhibit almost equal metal selectivity (and thus display quite similar ΔG^4/29^ in Figure 1a–c and upper part of Figure 1d).

Data analysis suggests that several factors—such as the number and type of protein ligands, charge state of the binding pocket and its solvent exposure—do not seem to play any significant role in governing the competition between Ca^2+^ and Sr^2+^ in CaSR. Rather, these are the intrinsic physicochemical properties of the two competing metal species that to a great extent orchestrate the process: Ca^2+^ has higher charge density than Sr^2+^ (0.40 vs. 0.27 e/Å^3^, respectively) and is a better Lewis acid than its bulkier counterpart. As a result, Ca^2+^ interacts more favorably with the protein ligands than Sr^2+^, yielding higher absolute value interaction energies, as shown in the first two rows of Table 1. Similar conclusions have been drawn for the competition between Ca^2+^ and Sr^2+^ in parvalbumin—a representative of the EF-hand family of proteins involved in calcium signaling [27]. CD and EF heptacoordinated binding sites, comprising Asp^−^, Glu^−^, Ser side chains, water and backbone peptide ligands, have been predicted to be Ca^2+^/Sr^2+^ selective. The conclusions have been confirmed by experimental measurements [27].

Although Sr^2+^ does not substitute for the native Ca^2+^ in CaSR, it is able to bind and activate the receptor (though with slightly lower efficacy than the native calcium) as it closely mimics the basic physicochemical and structural features of the native agonist. Note that the ΔG^4/29^ barrier of a few kcal/mol (Figure 1) could be easily overcome (thus turning the balance in favor of Sr^2+^) if the local concentration of Sr^2+^ increases within the bone microenvironment [24,29].

### 2.2. Fe^2+^ vs. Mg^2+^, Mn^2+^ and Zn^2+^ in Non-Heme Iron Proteins

Iron, a redox-active element with oxidation state alternating between +2 and +3 (and sometimes +4), plays a key role in a number of essential biological processes such as respiration, cell division, nitrogen fixation, oxygen transport, nucleotide synthesis, oxidant protection, gene regulation, and protein structure stabilization [30,31]. In mononuclear non-heme iron proteins, the metal cation is usually coordinated to His and Asp^−^/Glu^−^ side chains [32]. Typical Fe^2+^ binding site configuration is His_2_(Asp^−^/Glu^−^)_1_, designated as “2-His-1-carboxylate facial triad motif” [33], which has been found in a large group of iron dioxygenases, hydrolases, and synthases [33,34,35,36,37,38]. Other combinations between His and acidic residues exist as well: His_1_(Asp^−^/Glu^−^)_2_, His_2_(Asp^−^/Glu^−^)_2_ and His_3_(Asp^−^/Glu^−^)_1_ [32]. The coordination number of Fe^2+^ varies between 5 and 6 with water or substrate molecules supplementing the coordination sphere. Inside the cell, Fe^2+^ faces a competition from other biogenic metal species (i.e., Mg^2+^, Mn^2+^, and Zn^2+^) for binding the protein. Although these metal species are characterized with the same charge and similar ionic radii (R_Fe2+_ = 0.78 Å, R_Mg2+_ = 0.72 Å, R_Zn2+_ = 0.74 Å, and R_Mn2+_ = 0.83 Å for hexacoordinated cations [20]), they possess different ligand affinities (due mostly to varying charge accepting abilities), as reflected in the Irving−Williams series [39]:Mg^2+^ < Mn^2+^ < Fe^2+^ < Co^2+^ < Ni^2+^ < Cu^2+^ > Zn^2+^

Magnesium and manganese ions, positioned at the far left-hand side of the series, have weaker ligand affinities than Fe^2+^ (see Table 1 as well). Zinc cations, on the other side, are much stronger binders than their Fe^2+^ counterparts (Table 1) and form, as a rule, more stable complexes. Several outstanding questions arise:How does the Fe^2+^ binding site sequesters the “right” (native) cation from the cellular fluids and protect itself from attacks by other biogenic cations such as Mg^2+^, Mn^2+^, and Zn^2+^?What kind of selectivity strategies do iron binding sites employ toward metal cations having different ligand affinities and cytosolic concentrations?What are the key factors governing the metal selectivity in Fe^2+^ proteins?

These questions have been addressed in a modeling study employing a combined DFT/PCM approach [32]. Figure 2 depicts iron binding sites in the so-called “2-His-1-carboxylate facial triad motif” where the metal’s coordination number is either 5 (Figure 2a) or 6 (Figure 2b). The iron binding site, comprising another ligand combination of 3 His and one Asp^−^/Glu^−^ side chains, is represented in Figure 2c.

Results obtained reveal the following trends: (i) Mg^2+^ and Mn^2+^ are not able to dislodge Fe^2+^ from the respective binding sites, as evidenced by the positive free energies of metal substitution in both the gas phase and condensed media. This finding comes as no surprise in view of the weaker ligand affinities of Mg^2+^ and Mn^2+^ cations relative to those of the Fe^2+^ cation (Table 1). However, Mn^2+^—being closer in physicochemical properties to Fe^2+^ than Mg^2+^ to Fe^2+^ (Table 1)—is a more potent iron contender than Mg^2+^ (less positive free energies for the Fe^2+^ → Mn^2+^ exchange than for the Fe^2+^ → Mg^2+^ substitution; Figure 2). (ii) The Fe^2+^ binding sites, however, are ill protected against attacks by the rival Zn^2+^ cations, which form more stable complexes (Table 1) and are able to displace Fe^2+^ from the respective metal centers (negative ΔG values for the Fe^2+^ → Zn^2+^ exchange in Figure 2). (iii) Solvation does not appear to be a key determinant of the metal selectivity in these systems, as it weakly affects the free energies of metal substitution and does not alter the trends observed in the gas phase.

The theoretical results imply that Mg^2+^ cannot successfully compete with Fe^2+^ in these binding sites (relatively high positive free energies evaluated for the Fe^2+^ → Mg^2+^ substitution; Figure 2). The major determinant of the high Fe^2+^/Mg^2+^ selectivity in these systems is the chemical nature of the contending metal species which confers on Fe^2+^ higher ligand affinity than on Mg^2+^.

Divalent manganese and iron cations are neighbors in the Irving−Williams series, exhibiting similar ligand affinities (Table 1), ion radii (see above), coordination preferences (penta- or hexacoordinated first-shell ligand complexes), and cytosolic concentrations (in the micromolar range [1]), thus appearing to be comparably strong contenders for protein binding sites. The calculations show, not surprisingly, that iron centers, although still preferably binding Fe^2+^ (the latter being a better complexation agent than Mn^2+^), are weakly selective for Fe^2+^ over Mn^2+^ and are vulnerable to Mn^2+^ attacks. This is evidenced by positive, but low in absolute value, free energies (just a few kcal/mol) of the Fe^2+^ → Mn^2+^ exchange in both the gas phase and protein environment (Figure 2). The poor Fe^2+^/Mn^2+^ selectivity—supposedly resulting in the easily surmountable thermodynamic barrier for the Fe^2+^ → Mn^2+^ substitution—might, however, be advantageous for the cell metabolism and/or cell survival. Under conditions of Fe^2+^ deprivation, the iron protein may sequester Mn^2+^ cations from the surrounding fluids which, due to the close resemblance between the two metal species, might secure uninterrupted cell metabolism [40,41].

The zinc cation, characterized by greater ligand affinity than Fe^2+^ (Table 1), can outcompete the iron cation and displace it from its binding sites regardless of their composition, structure and solvent exposure (negative free energies for the Fe^2+^ → Zn^2+^ substitution in the entire series of complexes (Figure 2). The results are in line with a number of in vitro experiments showing that, indeed, Zn^2+^ binds to the host protein with much greater affinity than Fe^2+^ [42,43,44,45,46]. Note that, although the protein preferentially coordinates to Zn^2+^ in vitro, it binds and is activated by Fe^2+^ in vivo [42,43,44,45,46,47]. Inside the living cell, since the protein alone is not able to repel the attacks by the rival Zn^2+^, it is the cell machinery which, by tightly controlling the metal homeostasis and maintaining the free Zn^2+^ concentration at very low levels (in the picomolar to femtomolar range [1]), turns the balance in favor of Fe^2+^ whose free cytosolic concentration is in the micromolar range [1].

The theoretical study suggests that the inherent physicochemical properties of the contending metal species, reflected in the Irving−Williams series, are the major factor governing the metal selectivity in the non-heme iron centers. In addition, the free cytosolic concentration of the metal competitors, which correlates inversely with the Irving−Williams series, also affects the process of metal competition in vivo.

Notably, by using theoretical calculations, Kumar and Satpati have also found that the metal affinity of the wild-type and mutant CRISPR-associated protein 1 (with the first-shell E190, H254 and D268 residues lining the metal binding site) is in the order Ca^2+^ < Mg^2+^ < Mn^2+^ which is in agreement with the Irving−Williams series [48].

### 2.3. Competition between Cr^3+^ and Fe^3+^, Fe^2+^, Mg^2+^ and Zn^2+^ in Chromodulin

Chromodulin (low molecular weight chromium-binding substance, LMWCr) is a 1.5 kDa oligopeptide that, in Cr^3+^-loaded form, plays an essential role in the metabolism of carbohydrates and lipids by interfering with the insulin signaling pathways [49]. It has been implicated in reducing the insulin resistance in type 2 diabetic patients [50,51]. Although chromodulin’s primary and 3D structure have not yet been unraveled, it is known that chromodulin contains only four amino acid types in the ratio of Glu^−^:Gly:Cys:Asp^−^ = 4:2:2:2 [52]. An indispensable integral part of the oligopeptide in its active (holo-) form are four chromium cations in the oxidation state of 3+, located in two metal binding sites containing three and one Cr^3+^ ions (“3 + 1” mode of binding). Structural investigations on holo-chromodulin are not abundant and, to date, only limited information is available about the basic characteristics of the Cr^3+^ binding sites. The seminal experimental study of Jacquamet et al. sheds light on the following aspects of the metal-occupied binding centers [53]: (i) chromium does not alter its oxidation state upon binding and forms Cr^3+^ complex with the host chromodulin; (ii) the four Cr^3+^ cations are clustered into two separated centers containing 3 and 1 metals; (iii) metal cations are six-coordinated surrounded by oxygen-containing ligands arranged in a nearly octahedral fashion; (iv) cysteine side chains, oligomer end groups as well as water molecules appear not to be likely ligands for the metal, nor have sulfur bridges involving cysteines been identified; (v) Cr^3+^ cations in the trinuclear center are organized in the form of a (not ideal) isosceles triangle with the shorter side intermetallic distance of ~2.79 Å and longer sides lengths of ~3.79 Å; (vi) the metal ion pair forming the shorter side of the triangle is “glued” by hydroxo (but not oxo) bridges, whereas Asp^−^/Glu^−^ carboxylate bridges connect these Cr^3+^ cations with the third, more distant Cr^3+^.

Note that the paradigm of Cr^3+^ binding to chromodulin is especially intriguing from both experimental and theoretical points of view, since LMWCr appears to be the only molecule of biochemical importance whose native metal cofactor is Cr^3+^. This prompts several questions: Why chromium? What are the advantages of binding Cr^3+^ over other cellular biogenic metal species (e.g., Fe^3+^, Fe^2+^, Mg^2+^, Zn^2+^)? What factors influence the metal cation competition in chromodulin? These questions have been addressed recently by modeling the holo-chromodulin binding sites (following closely the findings from the experimental structural studies mentioned above) and evaluating the free energies of metal competition between the native Cr^3+^ and other biologically relevant metal species such as Fe^3+^, Fe^2+^, Mg^2+^ and Zn^2+^ [54]. A combination of density functional theory (DFT) calculations and polarizable continuum method (PCM) computations has been employed.

Guidelines derived from the experiment (see above) have been used to model the structures of the mono- and trinuclear Cr^3+^ metal centers (Figure 3). The basic characteristics of the optimized metal complexes are in agreement with the available experimental data: (i) the Cr^3+^ cations are six-coordinated with oxygen-containing ligands (acetates, backbone amide groups and hydroxyls) orbiting the metals in an octahedral fashion; (ii) neither water nor sulfur-containing ligands coordinate the metal cations; (iii) in the trinuclear sites, the metal cations form an isosceles triangle with two Cr^3+^ from the shorter side being connected by hydroxo bridges, whereas the third Cr^3+^ is linked to them by acetate bridges and a hydroxo-bridge. Note that the calculated geometrical parameters of the triangle are in good agreement with those evaluated experimentally [53]: R_Cr–Cr_ shorter side (Calc) = 2.79–2.80 Å and R_Cr–Cr_ shorter side (Exp) = ~2.79 Å; R_Cr–Cr_ longer side (Calc) = 3.58–3.65 Å and R_Cr–Cr_ longer side (Exp) = ~3.79 Å.

The experiment does not provide information about the nature of the non-bridging oxygen-containing ligands. Therefore, several metal centers have been constructed comprising various combinations of non-bridging acetates and backbone amides while, at the same time, maintaining the triangle structure with the respective OH^−^ and acetate bridges (Figure 3). Figure 3A represents a trinuclear metal center with two bridging and three non-bridging acetates, two of the latter binding the metal monodentately and the other one coordinated in a bidentate fashion. Figure 3B depicts a metal construct with two non-bridging acetates and two backbone amides, whereas Figure 3C,D show structures with one non-bridging acetate and two backbone ligands, and four backbone amides, respectively. Differences in the non-bridging acetate/backbone surrounding of the metal cations, as seen from Figure 3, have little effect on the geometry of the triangle structure, which remains virtually unaltered throughout the group (very similar Cr–Cr bond distances in all the structures).

The respective mononuclear binding sites have been modeled in agreement with the composition of their partner trinuclear structures: since the number of carboxylic residues in the host oligopeptide is 6 (2Asp^−^ and 4Glu^−^, see above), the number of acetates in the mononuclear constructs has been adjusted to that in the trinuclear center so that the total number of carboxylates in the two binding sites (trinuclear and mononuclear) sums up to 6. Thus, for the mononuclear structure in Figure 3A, the number of acetates is 1, whereas for those in Figure 3B–D is 2, 3 and 4, respectively. The rest of the metal’s co-ordination sphere has been complemented by backbone amide and hydroxyl ligands which, along with the acetates, surround the metal octahedrally.

The Gibbs free energies for the Cr^3+^ → M^3+/2+^ (M = Fe, Mg, Zn) metal substitution in chromodulin binding sites are also given in Figure 3. The data presented reveals that the trivalent Fe^3+^ cannot outcompete Cr^3+^ in either mononuclear or trinuclear metal centers, as demonstrated by positive ΔGs ranging from 2 to 7 kcal/mol for the former and between 14 and 20 kcal/mol for the latter. Evidently, the trinuclear chromium center is more resistant to Fe^3+^ attack than its mononuclear counterpart (higher positive ΔGs for the trinuclear structure compared to those for the mononuclear binding site). These findings are not unexpected in view of the higher affinity of Cr^3+^ to oxygen-containing ligands stabilizing the chromic complexes to a greater extent than the respective ferric structures. As seen from the data collected in Table 1 (bottom two rows), the energies of formation of chromic-single ligand complexes (ligand = species building metal binding sites in chromodulin, i.e., acetate, backbone amide and hydroxyl) are lower (more favorable) than their Fe^3+^ counterparts. Note, however, that the trend reverses for sulfur- (CH_3_S^−^) and nitrogen-containing (imidazole) amino-acid residues, which preferably bind Fe^3+^ over Cr^3+^. Importantly, such ligands, which would promote Fe^3+^ over Cr^3+^ selectivity, do not participate in metal binding in chromodulin, as they are either absent from the amino acid sequence (His) or, even though part of the oligopeptide buildup (Cys), are, apparently, far from the metal binding center [55].

Since the Lewis acidity/complexation power of divalent metals is lower than those of the trivalent cations (see Table 1), it is expected that M^2+^ metals would be weaker competitors to Cr^3+^. Indeed, as the numbers in Figure 3 suggest, Cr^3+^ binding sites are very well protected against attacks from divalent biogenic metals: ΔGs of the Cr^3+^ → M^2+^ (M = Fe, Mg and Zn) substitution vary between 127 and 165 kcal/mol in mononuclear complexes, and between 251 and 330 kcal/mol in the trinuclear constructs. The inherent physicochemical characteristics of the rival metal species emerge as the key factor controlling the metal competition in LMWCr.

Both experimental and theoretical findings point out at the unique role of Cr^3+^ as a chromodulin metal co-factor. Due to its high affinity toward oxygen-containing ligands (higher than those of its metal competitors; Table 1), Cr^3+^ binds very tightly to chromodulin, ensuring a robustly built complex. The apo-oligopeptide host—which, presumably, is quite flexible and therefore, so far, defies crystalization and X-ray examination—needs a potent immobilizer (i.e., Cr^3+^) in order to achieve its proper active fold, that is recognized by the insulin receptor. The chromic complex is not prone to attacks by other metal species from the cellular environment and remains structurally unaltered in their presence. Two major factors contribute to this: (1) The amino acid residues, such as Cys and His, which could have enhanced Fe^3+^ competitiveness (see Table 1) and thus compromised the structure and properties of the native holo-chromodulin, have been excluded (supposedly during the cell evolution) from coordinating the metal cofactor. (2) Since chromium’s 3+ oxidation state is preferred over its 2+ state (reduction potential for Cr^3+^/Cr^2+^ = −0.41 V [56]), the more stable Cr^3+^ complexes are expected to dominate over the alternative (and weaker) Cr^2+^ constructs in the cellular space. On the other hand, other potential trivalent competitors (Fe^3+^, Co^3+^ and Mn^3+^) exhibit preference for the 2+ oxidation state (reduction potentials for M^3+^/M^2+^ = 0.77, 1.60 and 1.93 V for Fe, Mn and Co, respectively [56]) which, in the form of M^2+^ metals, reduces their competitiveness toward Cr^3+^.

## 3. Metal Coordination Number Is an Important Determinant of the Metal Selectivity

Sodium (Na^+^) is an indispensable allosteric regulator in a number of signal-transducing proteins, such as neurotransmitter transporters and G-protein coupled receptors (GPCRs). These have been recognized as drug targets for psychiatric disorders [57,58] and addictive behavior [59]. In the holo-protein, sodium cation(s) are usually penta- or hexacoordinated and predominantly bind to oxygen-containing ligands such as Asp^−^/Glu^−^ and Ser/Thr side chains, backbone peptide groups or water [60]. The competition between Na^+^ and Li^+^ (non-biogenic metal cation known for its beneficial therapeutic effect on patients with mental disorders) in model sodium-binding sites have been studied by a combined DFT/PCM approach [60], and key determinants controlling the selectivity for Na^+^ over Li^+^ in sodium proteins have been elucidated.

Two types of sodium binding sites have been modeled: flexible ones that allow for ligand rearrangement upon Na^+^ → Li^+^ exchange (black numbers in Figure 4); and rigid binding pockets which preserve the original ligand arrangements in the “mother” sodium complex during metal substitution (blue numbers in parentheses in Figure 4). The results reveal that the coordination number of the competing metals is an important factor in the selectivity process: Li^+^, when allowed to adopt its preferred tetrahedral ligand arrangement (decreasing its coordination number from 6 to 4), outcompetes the six-coordinated Na^+^ in the entire dielectric range. On the other hand, in rigid binding sites, competitiveness of the lithium cation decreases (positive free energies in protein environment; blue numbers in parentheses) as it is forced to adopt the unfavorable octahedral ligand surrounding of the native sodium: thus, ligand repulsion between the six bulky ligands around the small Li^+^ cation attenuates its efficiency of binding. Note that the rigid binding sites preserve the original, relatively large, binding cavity optimized to fit the size of the bulkier Na^+^ but not the smaller Li^+^, which additionally decreases the strength of the interactions between Li^+^ and ligands lining the pore.

Furthermore, theoretical evaluations suggest that increasing the coordination number of the rival metal cation usually decreases its competitiveness. As the calculations demonstrate, increasing the coordination number of Sr^2+^ from 7 to 8 (by changing the binding mode of a carboxylate residue from mono- to bidentate as shown in Figure 1d, lower part) additionally weakens the strength of the electrostatic interactions in the Sr^2+^ complex (related to increased ligand repulsion around the metal cation) which, as seen, results in higher (less favorable) free energy of Ca^2+d^ → Sr^2+^ metal substitution (compare Figure 1d upper and lower parts).

## 4. Adjacent Metal Cation May Reverse the Metal Selectivity in Binuclear/Trinuclear Metal Binding Sites

Nickel-containing enzymes are of vital importance for a number of plants and primitive organisms, such as archaea, bacteria, fungi and low-trophic level marine eucaryota [61,62], where they fulfill various tasks ranging from energy generation to detoxification, oxidative stress protection and virulence [63]. To date, nine nickel-dependent biocatalysts, subdivided into non-redox (urease, glyoxylase I, acireductone dioxygenase and lactate racemase) and redox enzymes (CO-dehydrogenase, acetyl-CoA synthase, [NiFe]-hydrogenase, methyl-SCoM reductase, and Ni-superoxide dismutase), have been identified and characterized [64]. The structure and composition of their metal centers are quite diverse, varying from mononuclear nickel binding sites to homo-binuclear and hetero-binuclear constructs [62,63,64,65]. Cysteine is the predominant amino acid residue in redox enzymes, whereas histidines and aspartates/glutamates are the ligands of choice in the non-redox metaloproteins. Nickel’s valence state in the latter is 2+, while it alternates between 1+, 2+ and 3+ in the former.

Nickel enzymes can be deactivated/inhibited by other metal cations such as Zn^2+^ (in *Escherichia coli* glyoxalase I) [66] and Ag^+^ (in urease) [67,68,69]. The inhibition by Ag^+^ of urease (a pathogen in humans) is of high significance as it may have strong implications for pharmacology and medicine. The mechanism of silver antibacterial action in urease, however, is still not completely understood. Although the prevailing hypothesis postulates that Ag^+^ binds to some sulfur containing amino acid ligands (not nickel-bound) at the periphery of the active site which disrupts the enzyme structure [67], the substitution of Ni^2+^ cations from the metal center by Ag^+^ cannot be excluded [68,69]. Therefore, it is of special interest to determine to what extent the Ni^2+^ binding sites in urease (and other nickel enzymatic binding sites) are predisposed to Ni^2+^ → Ag^+^ substitution. Of note, information on the factors controlling the competition between the cognate Ni^2+^ and abiogenic (“alien”) Ag^+^ in biological systems is critically lacking.

To fill in the gap, the rivalry between Ni^2+^ and Ag^+^ in nickel enzymes has been studied by a combined DFT/PCM approach and key determinants of the process have been revealed [70].

In Figure 5, the fully optimized Ni^2+^ and Ag^+^ structures of two non-redox mononuclear nickel enzymes (glyoxalase I and acireductone dioxygenase) along with the Gibbs free energies of Ni^2+^ → Ag^+^ substitution are presented.

Glyoxalase I utilizes intracellular thiols to convert cytotoxic ketoaldehydes, such as methylglyoxal, into nontoxic D-hydroxy acids [64]. The enzyme contains a mononuclear nickel center where two histidine and two glutamate amino acid residues along with two water molecules surround the metal in an octahedral fashion (Figure 5A, left-hand side). Upon Ni^2+^ → Ag^+^ substitution, the binding site undergoes quite drastic structural changes resulting in a distorted tetrahedral silver complex with an acetate (model for the Glu^-^ side chain) and a water ligand transferred to the second coordination layer of the metal (Figure 5A, right-hand side). The free energy calculations suggest that the nickel active center is well protected against Ag^+^ attack and that the “alien” Ag^+^ cannot dislodge the native Ni^2+^ from its binding site as evidenced by highly positive ΔG^ε^s spanning the entire dielectric region. It is of note that the gas-phase exchange reaction, which is entirely dominated by electronic effects (being in favor of the divalent Ni^2+^), is characterized by quite high positive ΔG^1^s. These are attenuated in the condensed phase, where solvation effects favor to a greater extent the monovalent Ag^+^, yielding smaller (although still positive) free energies of metal exchange. This is in line with findings from a similar investigation [70] (not shown here), which demonstrates that increasing the number of charged/polar ligands, surrounding the metal (2 methylimidazoles and 2 acetates representing His and Glu^-^ amino acid side chains, respectively) and donating more charge to divalent Ni^2+^ than to its monovalent contender, increases the competitiveness of Ni^2+^ over Ag^+^.

Acireductone dioxygenase is a mononuclear nickel enzyme involved in the methionine salvage pathway. In the process, methylthioadenosin is transformed into acireductone which, consequently, is converted to formate, carbon monoxide, and methylthiobutyric acid [64]. The nickel binding site comprises three histidines, one glutamate and two water molecules octahedrally arranged around the Ni^2+^ cation (Figure 5B left-hand side). Upon Ni^2+^ → Ag^+^ exchange, the complex isomerizes to a four-coordinated structure with a water molecule and methylimidazole ligand relegated to the metal second coordination sphere (Figure 5B, right-hand side). The positive ΔG^ε^s evaluated for the metal substitution reaction suggest that Ag^+^ cannot outcompete the native Ni^2+^ in this system (due mainly to the presence of strong charge-donating ligands in the binding cavity) and that the acireductone dioxygenase binding site is reliably shielded from “alien” monocationic attack.

Urease, a nickel-dependent enzyme, catalyzes the hydrolytic decomposition of urea producing ammonia and carbamic acid which, subsequently, decomposes into another molecule of ammonia and carbonic acid [63]. The enzyme has been recognized as a virulence factor in several pathogenic (mostly antibiotic-resistant) bacteria, which makes it a plausible target for antibacterial therapy [67,68,69]. Urease comprises a homo-binuclear active center where the protein donates four histidines (two to each metal), an aspartate (in binding site 2) and a bridging carbamylated lysine to the metal cations. The optimized structure of the Ni1-Ni2 binding site (following the X-ray data from 1FWJ) is shown in Figure 6.

The silver-substituted Ag1-Ni2 and Ni1-Ag2 constructs along with the Ag1-Ag2 complex are also given in Figure 6. Incorporating Ag^+^ in the binding sites, as expected, alters their structure (the silver cation prefers smaller coordination numbers and longer metal-ligand bond distances), although, as shown, in different fashion. Binding site 1 preserves its pentacoordinated structure but alters its shape from almost regular squire pyramidal construct (with the native Ni^2+^) to a distorted squire pyramidal one (with the “alien” Ag^+^). Moreover, the Ag1-ligand coordinative bonds are considerably elongated in comparison with the respective Ni1-ligand counterparts: the mean of Ag1-ligand bond distance is 2.488 Å, whereas that of the Ni1-ligand bond distance is 2.068 Å. The binding site 2 structure changes more dramatically upon Ni^2+^ → Ag^+^ exchange: it transforms from a hexacoordinated (nearly octahedral) complex to a semi-squire planar complex with two water molecules transferred to the second coordination layer of Ag^+^. The Ag2-ligand bond distances increase in length as well (the mean value for the Ag2-ligand is 2.485 Å whereas that of the Ni2-ligand is 2.134 Å).

Thermodynamic evaluations reveal that the first Ag^+^ → Ni^2+^ exchange in buried binding pockets (ε = 4) in either site is favorable, and characterized with negative ΔG^4^s (−4.2 kcal mol^−1^ for binding site 1, and −2.7 kcal mol^−1^ for binding site 2; Figure 6, left-hand side). Substituting the second Ni^2+^ cation with Ag^+^, however, is thermodynamically unfavorable, as Gibbs free energies for the entire dielectric region are positive (Figure 6, right-hand side). Notably, the calculations suggest that only one mole of metal cations is exchanged during the process.

Why is the Ag^+^ → Ni^2+^ substitution in binuclear urease more favorable than that in the mononuclear glyoxalase I and acireductone dioxygenase (see above)? Why is the competitiveness of Ni^2+^ over Ag^+^ in bimetallic center compromised? This is mainly due to the presence of a second neighboring metal atom in the active center which, with its positive charge, attenuates the strength of the charge transfer through the bridging carboxylate to the other metal. Indeed, instead of coordinating to the “pure” strong charge-donating anionic carbamylated lysine (represented by CH_3_CH_2_NHCOO^−^), the Ni1^2+^ cation, in fact, binds to the metal-bound cationic [CH_3_CH_2_NHCOO^−^-Ni2^2+^]^+^ ligand characterized with a poorer charge-donating power that significantly attenuates the strength of the interaction between the Ni1^2+^ and [CH_3_CH_2_NHCOO-Ni2^2+^]^+^. Note that, in [CH_3_CH_2_NHCOO-Ni2^2+^]^+^, the Ni2^2+^ cation withdraws electron charge density toward itself, thus reducing the amount of charge that the metal-bound carbamylated lysine can donate to the adjacent Ni1^2+^. As a result of the decreased charge-donating strength of the metal-bound carbamylated lysine, the Ni1^2+^–ligand interactions are attenuated to a greater extent than those of the Ag1^+^–ligand interactions which, as expected, results in reduced Ni1^2+^/Ag1^+^ competitiveness (witnessed by the results in Figure 6). The same considerations hold for the Ni2^2+^/Ag2^+^ exchange as well. At the same time, the donating power of the metal ligands in the mononuclear glyoxalase I and acireductone dioxygenase is not compromised, and Ni^2+^ remains the metal of choice for these enzymes. Note that similar conclusions, due to the same reasons, have been reached for other multinuclear metal banding sites where the generally weaker monovalent Li^+^ can outcompete the stronger divalent Mg^2+^ (in GSK-3β and inositol monophosphatase [71]) and Ca^2+^ (in protein kinase C [72]).

As mentioned, the theoretical study described above has found that the Ni^2+^ centers in the mononuclear glyoxalase I and acireductone dioxygenase are well protected against Ag^+^ attack (Figure 5), and, presumably, still functional in its presence. In fact, literature data on the Ni^2+^ → Ag^+^ exchange in these systems is lacking, pointing to a possible scenario whereby such a metal substitution in these bacterial enzymes is not likely.

On the other hand, the binuclear nickel binding site in urease, in accord with experimental findings [67,68,69], appears predisposed to silver attack (Figure 6). Although the dominant hypothesis on the mechanism of Ag^+^ antibacterial effect in urease contends that the effect stems mostly from altering the structure of some sulfur-containing domains around the active center by *2 moles* of Ag^+^ [67], the theoretical findings presented above suggest an alternative pathway incorporating a direct Ni^2+^ → Ag^+^ metal exchange in the active site (Figure 6) associated with significant structural changes there. Importantly, the finding that *1 mole* of metals is exchanged in the process is supported by the experiments performed by Ambrose on the jack bean urease, suggesting that the enzyme inhibition is caused by *one* metal cation [68].

## 5. Rigid Binding Sites Adapted to the Specific Structural Requirements of the Native Metal Cation Enhance Its Competitiveness toward Other Metal Contenders

Figure 4 (above) demonstrates how the rigidity of the binding site affects the metal ion competition: inflexible binding sites optimized to fit the steric preferences of the cognate cation appear to have a strong protective effect against attacking metal species of varying origin. The intruding small Li^+^ cation (ionic radius of 0.76 Å [20]) only loosely binds to the relatively large host protein cavity adapted to cradle the bulkier native Na^+^ cation (ionic radius of 1.02 Å [20]). This reflects on the free energy of the Na^+^ → Li^+^ exchange in the protein environment, which stays firmly on a positive ground (ΔG^4/10^ in parentheses in Figure 4) suggesting Na^+^-selective binding sites.

Rigid binding sites can effectively discriminate between the native and larger attacking cations as well. As shown in Figure 7, both CD and EF binding sites of parvalbumin, when inflexible, enhance the Ca^2+^/Sr^2+^ selectivity evidenced by greater positive numbers (in parentheses) evaluated for rigid binding sites relative to those of the fully flexible counterparts (not bracketed numbers) [27]. The Ca^2+^-adapted binding cavity (Ca^2+^ ionic radius = 1.06 Å [20]) squeezes the incoming bulkier Sr^2+^ cation (ionic radius = 1.21 Å [20]), thus preventing it from optimally coordinating the ligands lining the binding site.

## 6. pH of the Medium Is a Key Factor Governing the Competition between the Native and Trivalent “Alien” Metal Species

The non-biogenic aluminum (in a trivalent cationic form) has been implicated in some health disorders in humans: vitamin D-resistant osteomalacia, iron adequate microcytic anemia, amyotrophic lateral sclerosis, and Parkinson’s and Alzheimer’s disease [73]. Protein binding sites containing essential metals, such as magnesium, calcium or iron, have been identified as targets for the “alien” Al^3+^ [74]. Especially predisposed to Al^3+^ attacks are Mg^2+^ binding sites where the Mg^2+^ → Al^3+^ substitution appears to be one of the major mechanisms through which aluminum exerts its toxic effect [74]. Al^3+^ and Mg^2+^ share several common features which make the magnesium binding sites easily identifiable by the attacking Al^3+^ and promote the Mg^2+^ → Al^3+^ substitution: (i) both species are “hard” cations with preference for “hard” oxygen-containing ligands arranged octahedrally around the metal; And (ii) both are small cations with similar ionic radii: 0.54 and 0.72 Å for the six-coordinated Al^3+^ and Mg^2+^, respectively [20]. Surprisingly however, although aluminum is a non-biogenic metal, several plant and animal species tolerate quite well high doses of aluminum salts. Notably, the acute toxicity of aluminum in mammals is very high: the median lethal dose, LD_50_, for aluminum sulfate in mice, taken orally, is 6200 mg kg^−1^ [75], whereas that of another “alien” metal, Hg (in the form of HgCl_2_), is only 12.9 mg kg^−1^ [76].

Since Al^3+^ is a strong Lewis acid, it forms several types of hydrated species in aqueous solution with differing proportion of ionized (as OH^−^) and non-ionized (as H_2_O) water molecules, depending on the pH of the environment. Aluminum hexaaqua complex undergoes stepwise deprotonation characterized with four, narrowly spaced pK_a_ values: pK_a1_ = 5.5, pK_a2_ = 5.8, pK_a3_ = 6.0 and pK_a4_ = 6.2 [77]. Thus, in acidic solutions (pH < 5), the predominant species is the octahedral hexaaqua complex, [Al(H_2_O)_6_]^3+^. At ambient pH of 7, however, another soluble species prevails: the tetrahedral anionic {[Al(OH^−^)_4_](H_2_O)_2_}^−^ construct. Notably, almost all the soluble Al^3+^ at pH ~7 exists in the latter form as the molar ratio between {[Al(OH^−^)_4_](H_2_O)_2_}^−^ and [Al(H_2_O)_6_]^3+^ is 2.5 × 10^6^ [74]. In the circumstances, it is of particular interest to assess the potential of both aluminum species to influence the competition between the abiogenic Al^3+^ and native metal species for protein binding sites. To shed light on the intimate mechanism of the M^2+^ → Al^3+^ (M = Mg, Fe) metal substitution, DFT/PCM calculations have been performed and major determinants of the process have been unraveled [78].

Figure 8 depicts fully optimized metal-loaded structures of typical magnesium binding sites comprising three Asp^−^/Glu^−^ side chains (modeled as acetates; Figure 8A) and one Asp^−^/Glu^−^ residue and a backbone peptide group (represented by CH_3_CONHCH_3_; Figure 8B). The enthalpies and Gibbs free energies in condensed media for the metal exchange in acidic pH (below 5; red numbers) and neutral pH (~7; blue numbers) are also given. The calculations demonstrate that the aluminum complexation preserves the overall octahedral structure of the native magnesium complex and the relative position of the ligands. The metal-ligand bond distances in the “red” and “blue” aluminum complexes, however, vary depending on the protonation state of the metal-bound water ligands, which reflects on the overall charge of the complex (structural data not shown here). Thermodynamic parameters also vary with the nature of the incoming aluminum species: when the attacking species is [Al(H_2_O)_6_]^3+^ (acidic conditions) the substitution reaction Mg^2+^ → Al^3+^is favorable, especially in solvent-accessible binding sites (negative ΔG^29^s); when the charge (and, respectively, complexation power) on the aluminum cation in {[Al(OH^−^)_4_](H_2_O)_2_}^−^ is significantly reduced by the coordinating OH^−^ ligands, the metal substitution reaction is reversed—suggesting the magnesium binding sites are well protected against Al^3+^ attack at ambient pH (great positive “blue” numbers for both complexes in Figure 8). The same conclusions have been reached for the Fe^2+^ → Al^3+^ exchange reaction as well [78].

The results from the DFT/PCM calculations reveal that among the two major soluble Al^3+^ species, the [Al(H_2_O)_6_]^3+^ aqua complex is the one which is capable of substituting for the native divalent cation and, subsequently, inflicting some damage on the host metalloenzyme. On the other hand, however, the {[Al(OH^−^)_4_](H_2_O)_2_}^−^ species appear to be ineffective in dislodging the cognate metal from the enzymatic active centers: the competitiveness of the {[Al(OH^−^)_4_](H_2_O)_2_}^−^ species is considerably diminished by the high number of anionic hydroxyl ligands that reduce to a great extent the Al^3+^ charge-accepting ability and renders the electrostatic interactions with protein ligands not/less favorable.

Why is the toxicity of the Al^3+^ cation relatively low? The calculations shed light on this issue, as far as the mechanism of the metalloenzyme inhibition by the M^2+^ → Al^3+^ (M = Mg, Fe) substitution is concerned. The [Al(H_2_O)_6_]^3+^ species which is able to substitute for the native metal in several types of protein binding sites, is, in fact, present in very low (picomolar) concentrations at physiological pH of ~7 [74]. Thus, the portion of the aluminum-loaded active centers in proteins is expected to be quite low. The other soluble aluminum species at ambient pH, {[Al(OH^−^)_4_](H_2_O)_2_}^−^, on the other hand, can reach much higher concentrations in the cellular fluids (several μM [74]), but is thermodynamically incapable of displacing the cognate metal cation from the enzymatic active center (“blue” reactions in Figure 8). Thus, the combination between concentration and physicochemical factors renders the “toxic” M^2+^ → Al^3+^ (M = Mg, Fe) exchange a low-occurring event at physiological pH.

## 7. Mechanical Forces Can Modulate the Metal Selectivity in Metal Binding Sites in Proteins

Proteins are dynamic supple objects whose structure and biochemical properties are sensitive to the action of intra- and extracellular forces of varying origin. Applied mechanical forces play a substantial role in essential biological processes, such as cell growth, division, deformation, adhesion and migration [79,80]. The role of the applied mechanical stimuli in modulating the proteins’ performance is still not completely understood. For example, it is not clear to what extent mechanical forces may affect the metal affinity and selectivity of a metal binding site in metalloproteins. Moreover, it is not known whether mechanical forces with varying strength and directionality could be used as a tool to manipulate the metal binding properties of the metal center. This question has been addressed recently by employing M06-2X/6-311++G**//PCM calculations where the effect of applied mechanical stimuli on the competition between Ca^2+^ and Mg^2+^ in parvalbumin binding site has been studied [81]. Metal-loaded canonical EF-hand binding sites have been modeled and subsequently subjected to the action of mechanical forces with differing magnitude and directionality. Results are summarized in Figure 9.

The EF-hand-motif metal binding sites in resting state (no external forces) are, in agreement with the experimental findings, selective for Ca^2+^ over Mg^2+^ (positive free energies of metal exchange in both low and higher dielectric media; Figure 9A). Applying mechanical force along one of the metal-aspartate bonds (M-Asp1) alters the metal selectivity of the binding site which becomes almost non-selective when subjected to a pulling force of 1.01 nN (Figure 9B) but reverses its selectivity in favor of Mg^2+^ at greater applied force of 1.23 nN (negative ∆G^4^ of −2.2 kcal/mol and ∆G^10^ of −0.6 kcal/mol in Figure 9C). This is mostly because the Mg-Asp1 bond, being stronger, is more resistant to mechanical stress and deforms less at a given force than its Ca-Asp1 equivalent: 1.01nN force (Figure 9B) results in 0.127 Å elongation of the former bond whereas it stretches the latter by 0.2 Å. For the stronger force of 1.23 nN (Figure 9C), the respective numbers are 0.177 Å for the Mg-Asp1 bond and 0.3 Å for the Ca-Asp1 bond. These alterations in bond lengths reflect on the magnitude of the overall energy of the metal complexes which increases (meaning less favorable) a little for the Mg^2+^ construct (by 1.07 and 1.89 kcal/mol for 1.01 and 1.23 nN forces, respectively) but more significantly for the Ca^2+^ complex (1.75 and 3.34 kcal/mol for 1.01 and 1.23 nN forces, respectively). This shifts the metal-exchange reaction [Mg^2+^-aq] + [Ca^2+^ -protein] → [Mg^2+^-protein] + [Ca^2+^-aq] in favor of Mg^2+^. On the other hand, stretching the metal-backbone bond (M-Bkb) under the influence of a 0.25 nN force has an opposite effect, this time inflicting less significant structural changes in the Ca^2+^ complex than to its Mg^2+^ counterpart. This enhances Ca^2+^ competitiveness and increases the Ca^2+^/Mg^2+^ selectivity (increased positive ∆G^4^/∆G^10^ from 0.9/0.5 kcal/mol in Figure 9A to 2.1/2.9 kcal/mol in Figure 9D).

Using an EF-hand motif binding site as an example, the calculations suggest that applying mechanical forces with a given strength and directionality can modulate the metal affinity and selectivity of the binding site: forces directed mostly along strong, less deformable bond(s) formed by a given metal cation enhance its selectivity over its contender.

## 8. Concluding Remarks

Theoretical findings, presented here, demonstrate that high level QM/PCM modeling studies can furnish valuable information about the intimate mechanism (with atomic resolution) of metal binding and selectivity in metalloproteins. The trends of changes in thermodynamic quantities, being in line with available experimental data, help elucidate with high reliability the key factors governing these processes. Thus, inherent properties of the competing metal species (ionic size and charge, Lewis acidity, charge accepting ability, specific ligand affinity) are of crucial importance for the metal’s competitiveness. As demonstrated, in many cases these factors dominate over the other determinants of metal selectivity. It is important to note that, although this is usually the case with transition metals with strong complexation power (Zn^2+^ and Cu^2+^) which in in vitro experiments are predicted to outcompete weaker metal cations (Mg^2+^ and Fe^2+^), these metals are kept at bay (at very low concentrations) in in vivo conditions by the cell machinery in order not to interfere with the biochemistry of the weaker metal counterparts. In binuclear/trinuclear Ni^2+^, Mg^2+^ or Ca^2+^ binding sites, the neighboring metal cation is able to enhance the competitiveness of the weaker monovalent metal species (i.e., Li^+^ or Ag^+^), substituting for the adjacent dication and thus allowing for certain Ni^2+^, Mg^2+^ or Ca^2+^ enzymes involved in some human pathogeneses to be inhibited/deactivated. Increasing the coordination number of a given metal cation usually has an unfavorable effect on its competitiveness as it strengthens the repulsive interactions between the ligands surrounding the metal and, generally, decreases the stability of the metal construct. Rigidity of the metal binding site is another factor which very often plays a dominant role in shaping the binding site selectivity. Rigid/inflexible binding sites that do not allow ligand reorganization upon metal exchange, thus preserving the “mother” cavity size and shape, always support the native metal cofactor binding with respect to that of the incoming “intruder”. Hardening the binding site is, in most cases, achieved by creating an elaborate hydrogen-bond network among protein ligands from several coordination layers around the active site. When dealing with highly charged metal cations of valence state 3+ or 4+, it is of crucial importance to pay attention to the pH of the medium as it strongly affects the balance between the ionized and non-ionized water ligands coordinated to the metal and, respectively, the total charge of the hydrated metal complex. As the calculations reveal, external mechanical forces with particular strength and directionality may modulate the metal selectivity of the protein binding site.

## Figures and Tables

**Figure 1 molecules-28-00249-f001:**
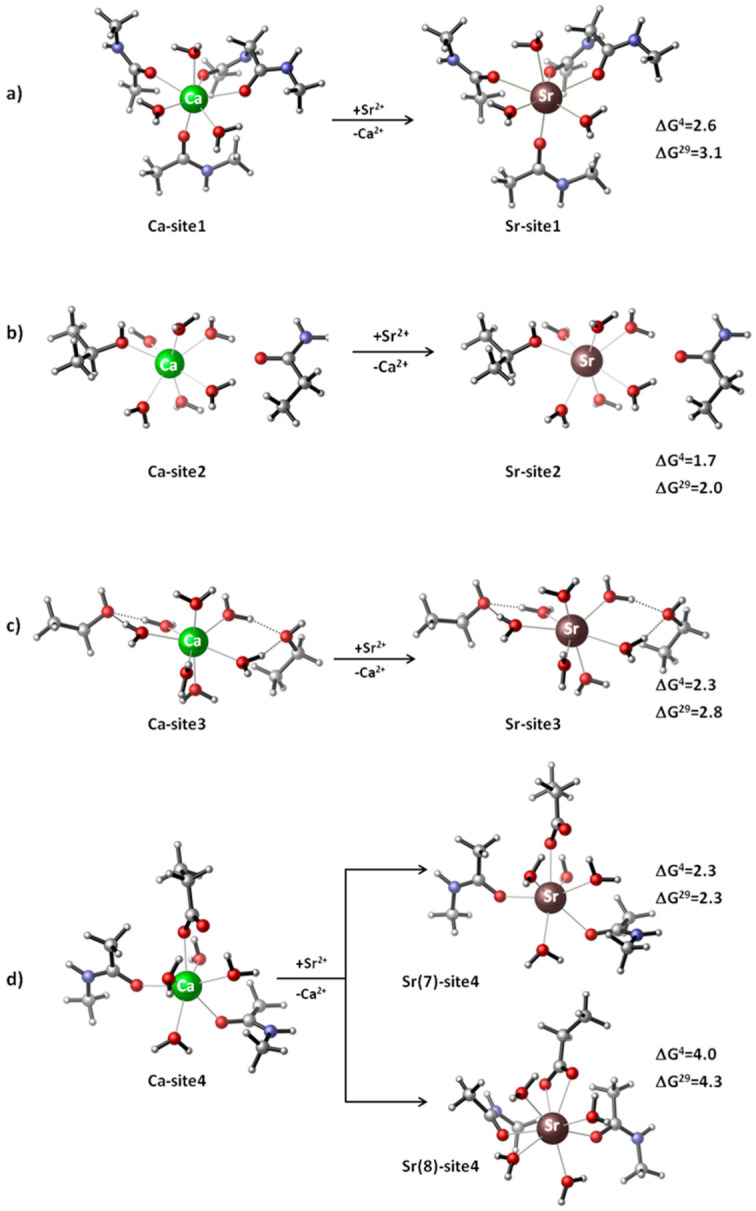
M06-2X/6-311++G**//SDD fully optimized Ca^2+^ and Sr^2+^-loaded metal binding sites of CaSR: (**a**) Site 1, (**b**) Site 2, (**c**) Site 3 and (**d**) Site 4. Gibbs free energies (in kcal/mol) of the Ca^2+^ → Sr^2+^ substitution are also given: ΔG^4^ stands for the free energy evaluated for buried metal binding sites, whereas ΔG^29^ signifies the metal exchange free energy for solvent-accessible binding pockets. Positive ΔG^4/29^ suggests a Ca^2+^-selective metal center. Reprinted from Ref. [28]. Copyright 2021 MDPI.

**Figure 2 molecules-28-00249-f002:**
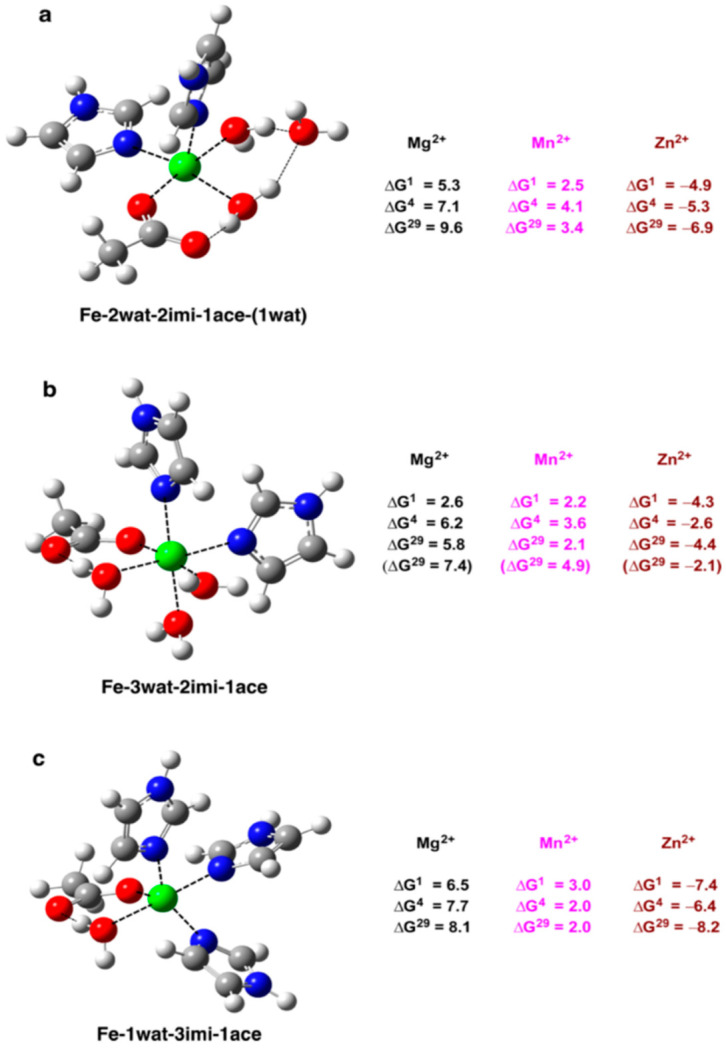
M06-2X/6-311++G** optimized structures of (**a**) pentacoordinated and (**b**) hexacoordinated Fe^2+^ model binding sites comprising two imidazole, one acetate, and three water ligands, and (**c**) pentacoordinated metal center containing three imidazole, one acetate, and one water ligands. The free energies ΔG^ε^ (in kcal/mol) for replacing Fe^2+^ in the binding site characterized by dielectric constant ε with M^2+^ (M = Mg, Mn, Zn) are shown on the right. ΔG^1^ refers to cation exchange free energy in the gas phase, whereas ΔG^4^ and ΔG^29^ refer to cation exchange free energies in an environment characterised by effective dielectric constants of 4 (buried binding sites) and 29 (solvent accessible binding pockets), respectively. Positive ΔG^ε^ suggests Fe^2+^-selective metal center, while negative ΔG^ε^ imply a M^2+^ selective site. His residues are represented by imidazoles (‘imi’) whereas the Asp^−^/Glu^−^ side chains are modeled as acetates (“ace”). Numbers in parentheses in Figure 2b refer to metal ion exchange in rigid Fe^2+^binding sites. Adapted from Ref. [32]. Copyright 2016 American Chemical Society.

**Figure 3 molecules-28-00249-f003:**
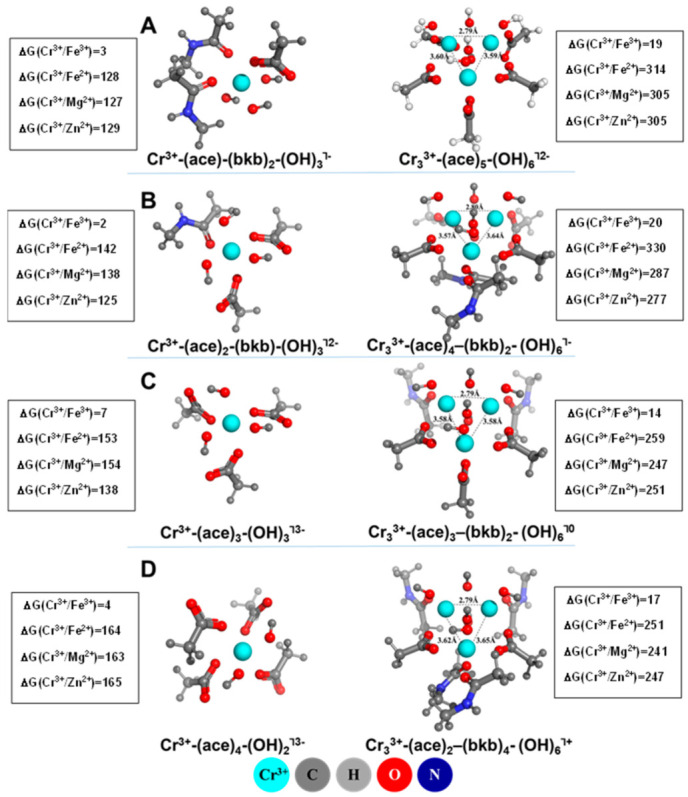
M06-2X/6-311++G** fully optimized structures of the Cr^3+^-loaded mono- and trinuclear chromodulin centers, and Gibbs free energies of Cr^3+^ → M^3+/2+^ (M = Fe, Mg, Zn) substitution (in kcal/mol) in relatively solvent exposed oligomer binding sites characterized with dielectric constant of 29. (**A**) number of acetates in the mono- and trinuclear binding sites is 1:5; (**B**) number of acetates in mono- and trinuclear binding sites is 2:4; (**C**) number of acetates in mono- and trinuclear binding sites is 3:3; (**D)** number of acetates in mono- and trinuclear binding sites is 4:2. Asp^−^/Glu^−^ side chains are represented by acetates (“ace”), CH_3_COO^-^, whereas the backbone peptide groups (“bkb”) are modeled as N-methylacetamide (CH_3_CONHCH_3_). Positive free energies imply a Cr^3+^-selective binding site. Adapted from Ref. [54]. Copyright 2022 Oxford University Press.

**Figure 4 molecules-28-00249-f004:**
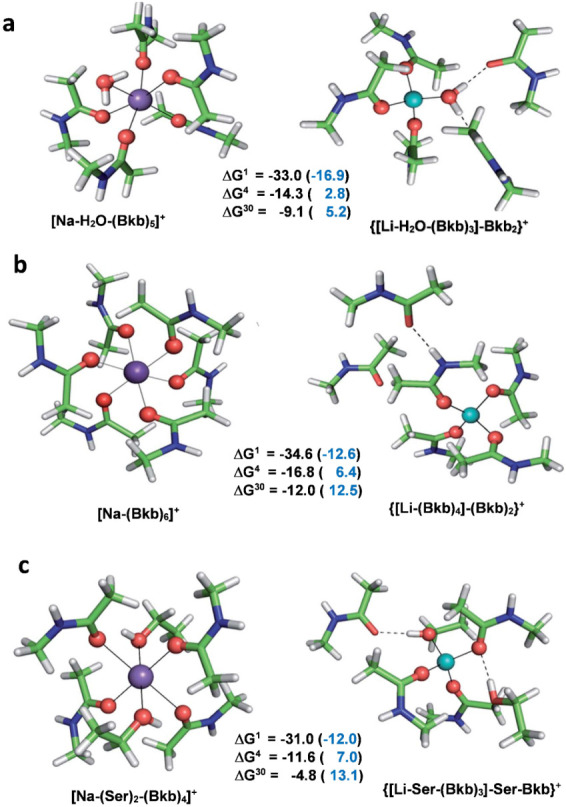
Calculated Gibbs free energies for the Na^+^ → Li^+^ exchange (in kcal/mol) and B3-LYP/6-31+G(3d,p) fully optimized structures of Na^+^ (purple) and Li^+^ (turquoise) complexes with (**a**) a water molecule and 5 backbone ligands; (**b**) 6 backbone ligands, and (**c**) 4 backbone and 2 serine ligands. ∆G^1^ refers to cation exchange free energy in the gas phase, whereas ∆G^4^ and ∆G^29^ refer to cation exchange free energies in an environment characterized by an effective dielectric constant of 4 and 29, respectively. Free energies for the metal exchange in rigid binding sites are given in parentheses. Positive numbers suggest Na^+^-selective sites, whereas negative free energies imply the opposite. Backbone peptide groups are modeled as CH_3_CONHCH_3_ and the side chains of Ser are represented as CH_3_CH_2_OH. Adapted from Ref. [60]. Copyright 2018 Royal Society of Chemistry.

**Figure 5 molecules-28-00249-f005:**
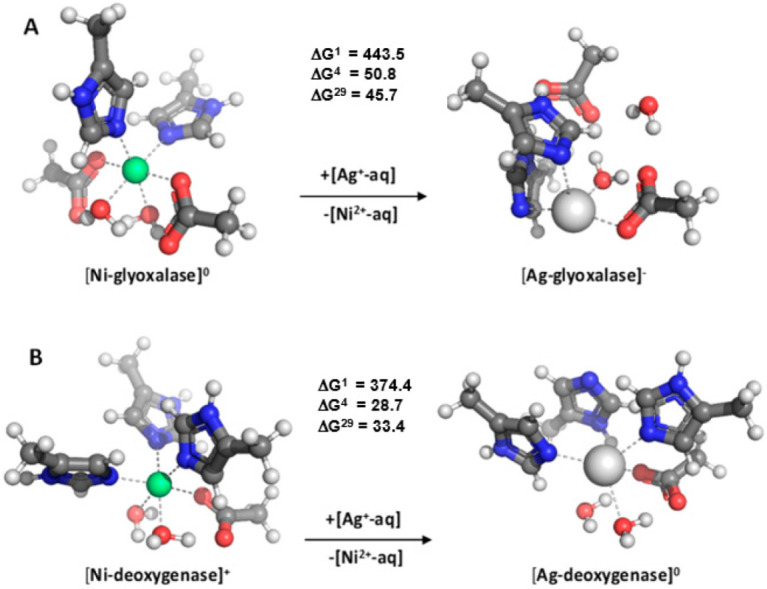
M06-2X/6-311++G** fully optimized glyoxalase (**A**) and acireductone dioxygenase (**B**) binding sites. The free energies ∆G^ε^ (in kcal mol^−1^) for replacing Ni^2+^ cation in the binding site characterized by dielectric constant ε with Ag^+^ ion are shown in the middle. ∆G^1^ refers to cation exchange free energy in the gas phase, whereas ∆G^4^ and ∆G^29^ refer to cation exchange free energies in an environment characterized by an effective dielectric constant of 4 and 29, respectively. Positive ∆Gs imply Ni^2+^ selective binding sites. Adapted from Ref. [70]. Copyright 2022 Elsevier.

**Figure 6 molecules-28-00249-f006:**
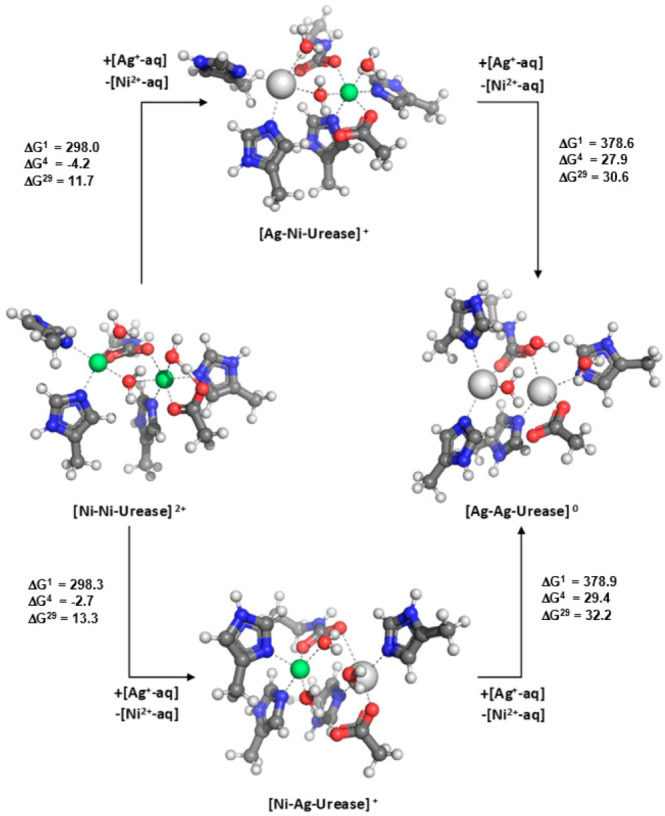
M06-2X/6-311++G** fully optimized structures of Ni^2+^/Ag^+^-loaded urease binding sites along with the Gibbs free energies, ∆G^ε^ (in kcal mol^−1^), for replacing Ni^2+^ with Ag^+^ in the binding sites characterized by dielectric constant ε. ∆G^1^ refers to cation exchange free energy in the gas phase, whereas ∆G^4^ and ∆G^29^ refer to cation exchange free energies in an environment characterized by an effective dielectric constant of 4 and 29, respectively. Positive ∆Gs suggest Ni^2+^-selective sites whereas negative free energies imply Ag^+^-selective constructs. Ni^2+^ and Ag^+^ are represented by green and grey spheres, respectively. Bridging carbamylated lysine is modeled by CH_3_CH_2_NHCOO^−^. Adapted from Ref. [70]. Copyright 2022 Elsevier.

**Figure 7 molecules-28-00249-f007:**
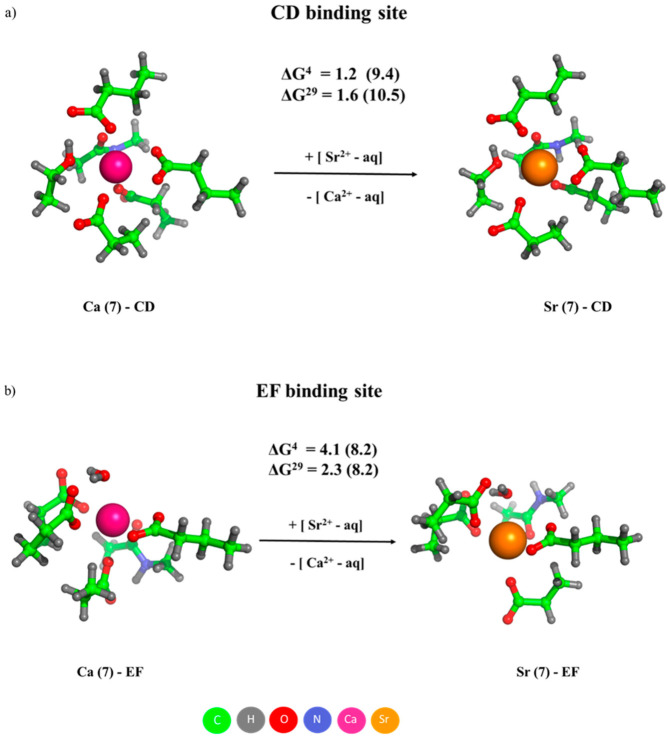
M06-2X/6-311++G** optimized structures of Ca^2+^/Sr^2+^-loaded CD (**a**) and EF (**b**) sites of parvalbumin, and free energies of Ca^2+^ → Sr^2+^ exchange (in kcal/mol). The free energies evaluated for rigid Ca^2+^-binding sites are given in parentheses. ∆G^4^ and ∆G^29^ refer to cation exchange free energies in an environment characterized by an effective dielectric constant of 4 and 29, respectively. Positive free energies imply Ca^2+^-selective binding sites. Reprinted from Ref. [27]. Copyright 2021 MDPI.

**Figure 8 molecules-28-00249-f008:**
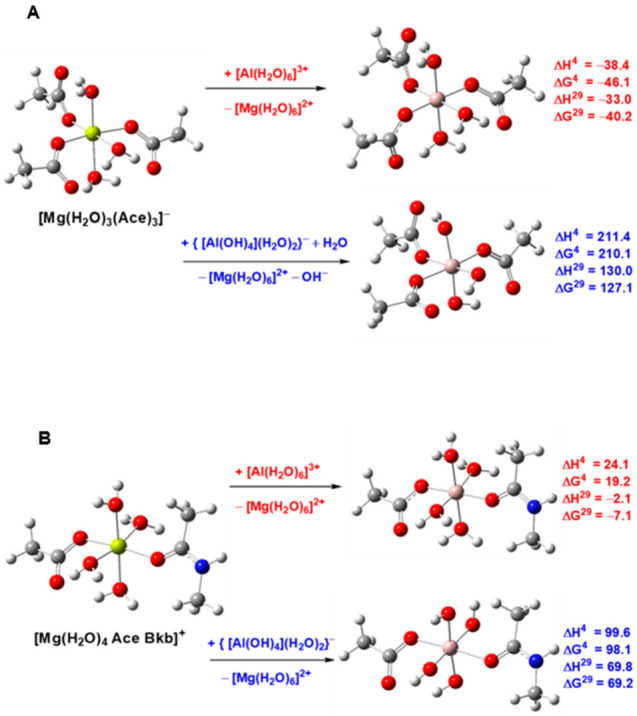
M06-2X/6-311++G** optimized structures of Mg^2+^ model binding sites comprising (**A**) three acetates and three water ligands, and (**B**) one acetate, one N-methylacetamide and four water ligands, and the respective resultant Al^3+^-containing structures obtained via Mg^2+^ → [Al(H_2_O)_6_]^3+^ and Mg^2+^ → {[Al(OH^−^)_4_](H_2_O)_2_}^−^ substitution. The enthalpies, Δ*H*^ε^, and Gibbs free energies, Δ*G*^ε^ (in kcal/mol), for replacing Mg^2+^ with Al^3+^ in the binding site characterized by dielectric constant *ε* are shown on the right and colored in red for the Mg^2+^ → [Al(H_2_O)_6_]^3+^ reaction, and in blue for the Mg^2+^ → {[Al(OH^−^)_4_](H_2_O)_2_}^−^ reaction. ΔH^4^/ΔG^4^ and ΔH^29^/ΔG^29^ refer to cation exchange enthalpy/free energy in an environment characterized by an effective dielectric constant of 4 and 29, respectively. Positive free energies imply a magnesium-selective binding site, whereas negative free energies suggest otherwise. Color scheme—Mg: yellow, Al: pink, O: red, N: blue, C: grey, H: slight grey. Adapted from Ref. [78]. Copyright 2018 Springer.

**Figure 9 molecules-28-00249-f009:**
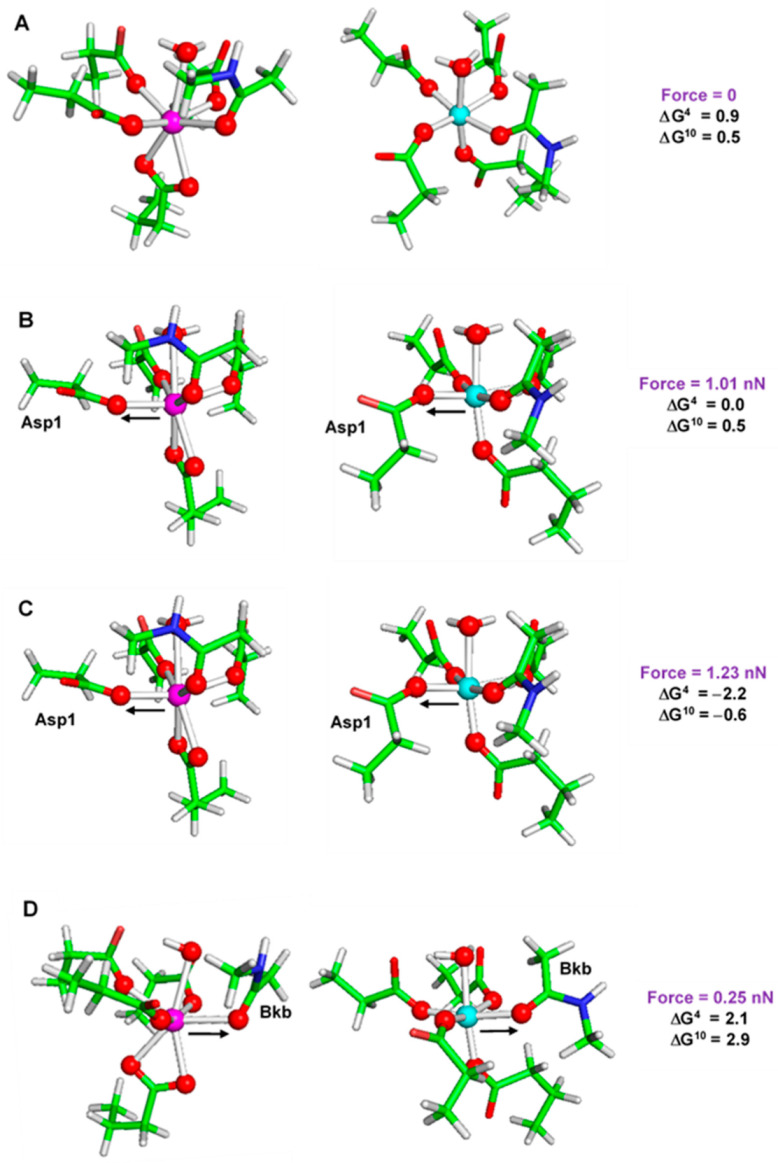
Free energies of Ca^2+^ → Mg^2+^ exchange (in kcal/mol) in metal complexes with external force of (**A**) 0 nN, (**B**) 1.01 nN applied along the M-Asp1 bond, (**C**) 1.23 nN applied along the M-Asp1 bond, and (**D**) 0.25 nN applied along the M-Bib bond. Positive ∆G^4^/∆G^10^ (for buried and partially buried binding sites, respectively) imply a calcium-selective binding site, whereas negative free energies signifies a magnesium-selective one. Abbreviations: nN = nano-Newton, Asp = aspartate, and Bkb = backbone peptide group. Col-or scheme—C: green, O: red, N: blue, H: light grey, Ca: magenta, Mg: cyan. Reprinted from Ref. [81]. Copyright 2020 Royal Society of Chemistry.

**Table 1 molecules-28-00249-t001:** Gas-phase energies of formation (in kcal/mol) of single-metal ligand complexes (metal = Ca^2+^, Sr^2+^, Fe^2+^, Mg^2+^, Mn^2+^, Zn^2+^, Cr^3+^, Fe^3+^) from M06-2X/6-311++G** calculations.

Metal	Ligand				
	CH_3_COO^− a^	CH_3_CONHCH_3_ ^b^	OH^−^	CH_3_S^− c^	Imidazole ^d^
Ca^2+^	−320.7	−108.3	−343.3	−292.6	−95.0
Sr^2+^	−297.9	−93.9	−321.3	−273.2	−81.8
Fe^2+^	−399.0	−163.0	−416.8	−388.1	−152.3
Mg^2+^	−375.1	−140.4	−383.6	−357.3	−134.7
Mn^2+^	−382.4	−147.8	−398.8	−372.1	−142.6
Zn^2+^	−414.0	−168.9	−417.5	−423.5	−174.1
Cr^3+^	−742.5	−440.6	−756.3	−770.4	−449.1
Fe^3+^	−740.0	−436.0	−736.0	−777.8	−453.6

^a^ Model for Asp^−^/Glu^−^ side chains; ^b^ Model for backbone peptide group; ^c^ Model for Cys^−^ side chain; ^d^ Model for His side chain.

## Data Availability

Research data are available by the author upon request.

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
