# Peer review of "How Theoretical Evaluations Can Generate Guidelines for Designing/Engineering Metalloproteins with Desired Metal Affinity and Selectivity"

_molecules, 2022, doi:10.3390/molecules28010249_

Round 1

Reviewer 1 Report

Comments to author:

 This review is dedicated to factors controlling the metal binding and selectivity in metalloproteins using as support high-level QM/PCM modelling studies. The thermodynamic properties obtained were used to elucidate on metal’s competitiveness. The work described is very well written and very clear. In my opinion, the article will be suitable for publication in Journal Molecules, after minor revision to improve some aspects.

The author should consider the following points:

1   1)      Legends 1 and 4 are incomplete. Please specify the delG types as in the legends 2 and 5.

2    2)      Figure 2b: three values of delG are in brackets. Some information should be added to the legend or in the text about it.

3    3)      Two misprints are found in the text: page 9, line 297 and page 19, line 578 (delG 4/30 instead of delG4/10?).

4    4)      Some studies have different metal exchange free energy for solvent-accessible binding pockets, namely dielectric constants of 29 or 30.  Is there any special reason for this? If so, please added this information in the text.

5    5)      In figures 5 and 6, the gas-phase delG values are significantly higher than for dielectric constants 4 or 30. This evidence occurs for a particular reason? Do you have some explanation for this aspect? One comment could be added to the text.

Author Response

Reviewer 1

The author should consider the following points:

1)      Legends 1 and 4 are incomplete. Please specify the delG types as in the legends 2 and 5.

Answer: The legends to Figures 1 and 4 are amended accordingly.

2)      Figure 2b: three values of delG are in brackets. Some information should be added to the legend or in the text about it.

     Answer: The information needed is provided.

3)      Two misprints are found in the text: page 9, line 297 and page 19, line 578 (delG 4/30 instead of delG4/10?).

     Answer: These are corrected.

4)      Some studies have different metal exchange free energy for solvent-accessible binding pockets, namely dielectric constants of 29 or 30.  Is there any special reason for this? If so, please added this information in the text.

      Answer: The correct number of the dielectric constant is 29. Dielectric constant of 30 was changed to 29 in Figures 4, 5 and 6.

5)      In figures 5 and 6, the gas-phase delG values are significantly higher than for dielectric constants 4 or 30. This evidence occurs for a particular reason? Do you have some explanation for this aspect? One comment could be added to the text.

     Answer: Few sentences were added in p. 17 to clarify the issue. 

Reviewer 2 Report

The manuscript is an interesting contribution, easy to read, and, in general, well built. I think that it can be acceptable in Molecules, but there are a few points that I would like to mention, in order of appearance in the text: 1.     Line: 81, 120-130, 164-166, 262-314, 630-641, 658-666: Wrong font and size used 2.     Image quality can be improved 3.     Some abbreviations are not universally known, for example line 54 post-HF and DFT 4.     It is also worth considering extending the introduction with an explanation of the DFT/PCM approach, which is mentioned many times in the text, 5.     Line: 797, 814, 892, 904: end reference page numbers are missing

Author Response

Reviewer 2

The manuscript is an interesting contribution, easy to read, and, in general, well built. I think that it can be acceptable in Molecules, but there are a few points that I would like to mention, in order of appearance in the text: 1.     Line: 81, 120-130, 164-166, 262-314, 630-641, 658-666: Wrong font and size used 2.     Image quality can be improved 3.     Some abbreviations are not universally known, for example line 54 post-HF and DFT 4.     It is also worth considering extending the introduction with an explanation of the DFT/PCM approach, which is mentioned many times in the text, 5.     Line: 797, 814, 892, 904: end reference page numbers are missing.

Answers: The size and type of fond in the pages, mentioned by the Reviewer, have been corrected.

The abbreviations mentioned in the text has been explained.

The introduction section has been amended and extended (blue text on p. 2)

The references, mentioned by the Reviewer, are correct. These are either single-page article (reference 30) or papers where the article number instead of page numbers is given (references 54, 70 and 78).